# Serine phosphorylation regulates the P-type potassium pump KdpFABC

**Marie E Sweet[1], Xihui Zhang[1], Hediye Erdjument-Bromage[1], Vikas Dubey[2], Himanshu Khandelia[2], Thomas A Neubert[1], Bjørn P Pedersen[3], David L Stokes[1]\***

[1]Skirball Institute, Dept. of Cell Biology, New York University School of Medicine, New York, United States; [2]PHYLIFE, Physical Life Science, Department of Physics, Chemistry and Pharmacy, University of Southern Denmark, Odense, Denmark; [3]Department of Molecular Biology and Genetics, Aarhus University, Aarhus, Denmark

**Abstract** KdpFABC is an ATP-dependent $K^+$ pump that ensures bacterial survival in $K^+$-deficient environments. Whereas transcriptional activation of kdpFABC expression is well studied, a mechanism for down-regulation when $K^+$ levels are restored has not been described. Here, we show that KdpFABC is inhibited when cells return to a $K^+$-rich environment. The mechanism of inhibition involves phosphorylation of Ser162 on KdpB, which can be reversed in vitro by treatment with serine phosphatase. Mutating Ser162 to Alanine produces constitutive activity, whereas the phosphomimetic Ser162Asp mutation inactivates the pump. Analyses of the transport cycle show that serine phosphorylation abolishes the $K^+$-dependence of ATP hydrolysis and blocks the catalytic cycle after formation of the aspartyl phosphate intermediate (E1~P). This regulatory mechanism is unique amongst P-type pumps and this study furthers our understanding of how bacteria control potassium homeostasis to maintain cell volume and osmotic potential.

**\*For correspondence:** stokes@nyu.edu

**Competing interests:** The authors declare that no competing interests exist.

## Introduction

Potassium is the primary osmolyte used by cells to maintain homeostasis and facilitate cell growth. A sizeable difference in $K^+$ concentration across the plasma membrane is largely responsible for setting the membrane potential, is essential for regulating intracellular pH, for generating turgor pressure required for cell growth and division and as an energy source for diverse transport processes (*Altendorf et al., 2009*; *Epstein, 2003*). When external $K^+$ concentrations are in the millimolar range, Trk and Kup are two constitutively expressed $K^+$ uptake systems that are sufficient to maintain $K^+$ homeostasis in bacteria. However, when extracellular $K^+$ concentrations fall into the micromolar range, a high-affinity, active transport system called Kdp takes over. The kdp genes, which underlie the only inducible $K^+$ transport system in bacteria, are organized into two tandem operons (kdpFABC and kdpDE). The latter is responsible for sensing the environment and controlling the expression of the former (*Jung and Altendorf, 2002*). The resulting KdpFABC protein complex is an ATP-dependent $K^+$ pump that has micromolar affinity for $K^+$ and is capable of generating a gradient up to six orders of magnitude, thus maintaining intracellular concentrations between 0.1 and 1M even when extracellular $K^+$ is present in trace amounts (*Ballal et al., 2007*).

Although the mechanisms for activating KdpFABC have been studied in considerable detail, the field has largely overlooked the need to suppress the pump once external $K^+$ levels return to the millimolar range. On the one hand, it is clear that activation of transport is done at the transcriptional level via KdpD/E, a two-component system in which KdpD serves as the sensor kinase and KdpE as the response regulator (*Walderhaug et al., 1992*). When the cell's need for $K^+$ is not being met, KdpD phosphorylates and thus activates KdpE, which then induces expression of the KdpFABC pump (*Jung et al., 2012*). The signal acting on KdpD is still controversial, with recent studies

providing evidence for external K⁺ concentration (*Laermann et al., 2013*), internal Na⁺ or NH₄⁺ concentrations (*Epstein, 2016*) or dual binding sites for K⁺ that sense the magnitude of the gradient across the membrane (*Schramke et al., 2016*). On the other hand, inhibition of transport by KdpFABC has been observed when cells are returned to K⁺-rich media (*Rhoads et al., 1978*; *Roe et al., 2000*). Inhibition of transport under these conditions is a logical necessity to prevent wasteful usage of ATP and a skyrocketing of the intracellular K⁺ concentration. Although elevated K⁺ levels do result in transcriptional repression, this is an unlikely explanation for the observed inhibition because the effect is far more rapid (<2 min in K⁺ rich media) than the documented rate of protein turnover in bacteria (*Trötschel et al., 2013*). Indeed, the authors reporting this effect speculated that existing KdpFABC molecules may be directly inhibited by some unreported mechanism (*Roe et al., 2000*).

Recent structural analysis revealed phosphorylation of a serine residue in a cytoplasmic domain of the KdpB subunit within a highly conserved sequence motif (*Huang et al., 2017*). This observation was surprising, given the prominence of this motif (TGES[162]) in the catalytic cycle of related P-type ATPases and the lack of precedent for an analogous post-translational modification in this well-studied superfamily of ATP-dependent cation pumps (*Palmgren and Nissen, 2011*). P-type ATPases including KdpB share a reaction cycle known as the Post-Albers scheme (*Figure 1*) that begins with formation of an aspartyl phosphate on a conserved sequence (D[307]KTGT) in the centrally located cytoplasmic domain (P-domain). This phosphoenzyme intermediate (E1~P) transiently harnesses the energy of ATP, which is then used to drive conformational changes that lead to ion transport. The highly conserved TGES motif is found in a second cytoplasmic domain (A-domain) with Glu161 playing a crucial role in hydrolysis of the aspartyl phosphate in the second half of the catalytic cycle (E2-P → E2) (*Møller et al., 2010*). Biochemically, The E1~P and E2-P states are distinguished by their reactivity to ADP, with the former, high-energy form rapidly dephosphorylating to produce ATP, whereas the latter, low-energy is unreactive (*Toustrup-Jensen et al., 2001*). An X-ray structure of KdpFABC showed that Ser162, which is immediately adjacent to this catalytic glutamate residue, carried a phosphate moiety and the ability of a protein phosphatase to increase ATPase activity suggested that phosphorylation of Ser162 was inhibitory (*Huang et al., 2017*). This crystal structure also revealed an interaction between the phosphate moiety and positively charged residues on the third cytoplasmic domain (N-domain), suggesting a possible mechanism for this inhibition. However, subsequent cryo-EM structures from another group which also showed phosphorylation of Ser162 (*Stock et al., 2018*), revealed different conformations indicating that the interaction between A- and N-domains was not stable and therefore probably not the basis for inhibition.

To further explore the inhibitory effects of serine phosphorylation, we have established growth conditions that lead to phosphorylation of Ser162 on KdpB and have characterized its effect on individual steps in the reaction cycle. Consistent with previous observations (*Roe et al., 2000*), when wild-type (WT) KdpFABC was expressed using the native promoter at low K⁺ concentrations, we found that moving cells to K⁺-rich media resulted in an inhibition of ATPase activity. We found that KdpFABC inhibition was time-dependent, was proportional to the extent of serine phosphorylation, and was restored by removing the phosphate with lambda protein phosphatase (LPP). To elucidate the mechanism of inhibition, we expressed a variety of mutants using the pBAD promoter in cells grown in K⁺-rich media and measured steady-state levels of the aspartyl phosphate intermediates (EP). We introduced the KdpB-S162D mutation to mimic the effect of serine phosphorylation and the KdpB-S162A

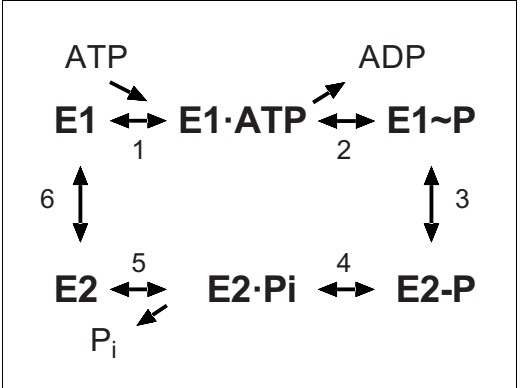

**Figure 1.** The Post-Albers scheme for the reaction cycle of P-type ATPases. The cycle alternates between two major conformations denoted E1 and E2. The E1 state binds ATP and produces the high-energy aspartyl phosphoenzyme intermediate (E1~P), which remains reactive with ADP. A spontaneous transition produces the low-energy E2-P intermediate, which no longer reacts with ADP but instead undergoes hydrolysis to produce the E2 species.

mutation to prevent serine phosphorylation. Despite a lack of inhibition with KdpB-S162A, we found that EP levels for this mutation were much lower than for serine-phosphorylated, WT KdpFABC or for the KdpB-S162D mutation. Pulse chase studies with ADP indicated that both serine phosphorylation and the KdpB-S162D mutation trapped the pump in the E1~P state. Furthermore, although EP formation with the KdpB-S162A mutant was K⁺-dependent, serine phosphorylation as well as the phosphomimetic S162D mutation eliminated this K⁺-dependence. These observations suggest that serine phosphorylation disrupts allosteric coupling of the pump and provides secondary level of control over K⁺ homeostasis.

## Results

### Expression of KdpFABC

We used three different expression strains and a variety of different growth conditions to study the physiological stimulus for serine phosphorylation and to produce KdpFABC for in vitro experiments (*Figure 2*, *Table 1*). In native *E. coli* cells, the promoter that controls transcription of kdpFABC genes is regulated by KdpE in response to 'potassium need' of the cell. In K⁺ replete conditions, KdpFABC is not expressed, but when K⁺ levels in the media fall into the micromolar range, the constitutive transport systems (Trk, Kup) fail to maintain the necessary chemo-osmotic gradient. KdpD senses this deficiency and activates KdpE such that transcription of kdpFABC is initiated. WT KdpFABC is a powerful pump, so the deficit is normally overcome by modest levels of expression. In order to obtain over expression, as required for structural or biochemical studies, a multicopy plasmid with kdpFABC driven by the native promoter is introduced into a cell line (TK2499) with Trk and Kup knocked out and cells are cultured with limiting concentrations of K⁺, which must remain in the low micromolar range for expression of WT kdpFABC (*Epstein and Davies, 1970*). Because KdpFABC activity is required to rescue these cells, mutations that generate lower activity cannot survive in these conditions and, in those cases, precise K⁺ concentrations that allow cell growth and also produce robust expression depend on the K⁺ affinity and/or turnover rate of the particular mutant (e.g. 0.2 mM for the KdpA-Q116R mutation used for the X-ray structure). Inactive mutants cannot be expressed using this strategy, because they are incapable of rescuing cell growth in a K⁺-deficient environment. In those cases, the pBAD promoter can be used for expression with Top10 cells in K⁺ replete conditions (e.g. for previous cryo-EM structures), or a chromosomal copy of the kdpFABC gene can be used for rescue (TK2498 cells) and the mutant protein can be expressed from a plasmid carrying the native promoter. In this case, a His-tag is used to separate the mutant from the chromosomal copy by affinity chromotagraphy. In the following studies, we have used all three expression strategies. As we learned over the course of this work, the K⁺ concentration in the culture media is key to serine phosphorylation, making the expression protocol an important parameter to keep in mind.

### Survey of KdpFABC mutants reveals unexpected inhibition

An overarching goal of our work is to understand the allosteric coupling between KdpA and KdpB that underlies ATP-dependent K⁺ transport. To this end, we setup a survey of alanine substitution mutants to assess the role of various residues implicated by the X-ray crystal structure. For expression of these mutants, we used *E. coli* strain TK2498 grown at 0.2 mM K⁺ (*Figure 2*, *Table 1*). After purification, the ATPase activities of all of the mutants were very low (*Figure 3a*), causing us to explore parameters that might have influenced the protein. Based on our earlier evidence that phosphorylation of Ser162 in KdpB is inhibitory (*Huang et al., 2017*), we used mass spectrometry to determine levels of this post-translational modification (*Figure 3—figure supplement 1*), which were all very high (*Figure 3b*). We then used LPP to remove the phosphate and observed significant stimulation in ATPase activity for several of these mutants (*Figure 3c*), thus supporting an inhibitory role for Ser162 phosphorylation. WT protein and the D307A mutation in KdpB served as positive and negative controls, respectively. As expected, the D307A mutant was catalytically inactive because this residue is the site of aspartyl phosphate (E1~P) formation. In contrast to the mutants, the WT construct had virtually no serine phosphorylation. This is because the WT protein has much higher K⁺ affinity and was therefore expressed at much lower levels of K⁺ (<10 µM) in the TK2499

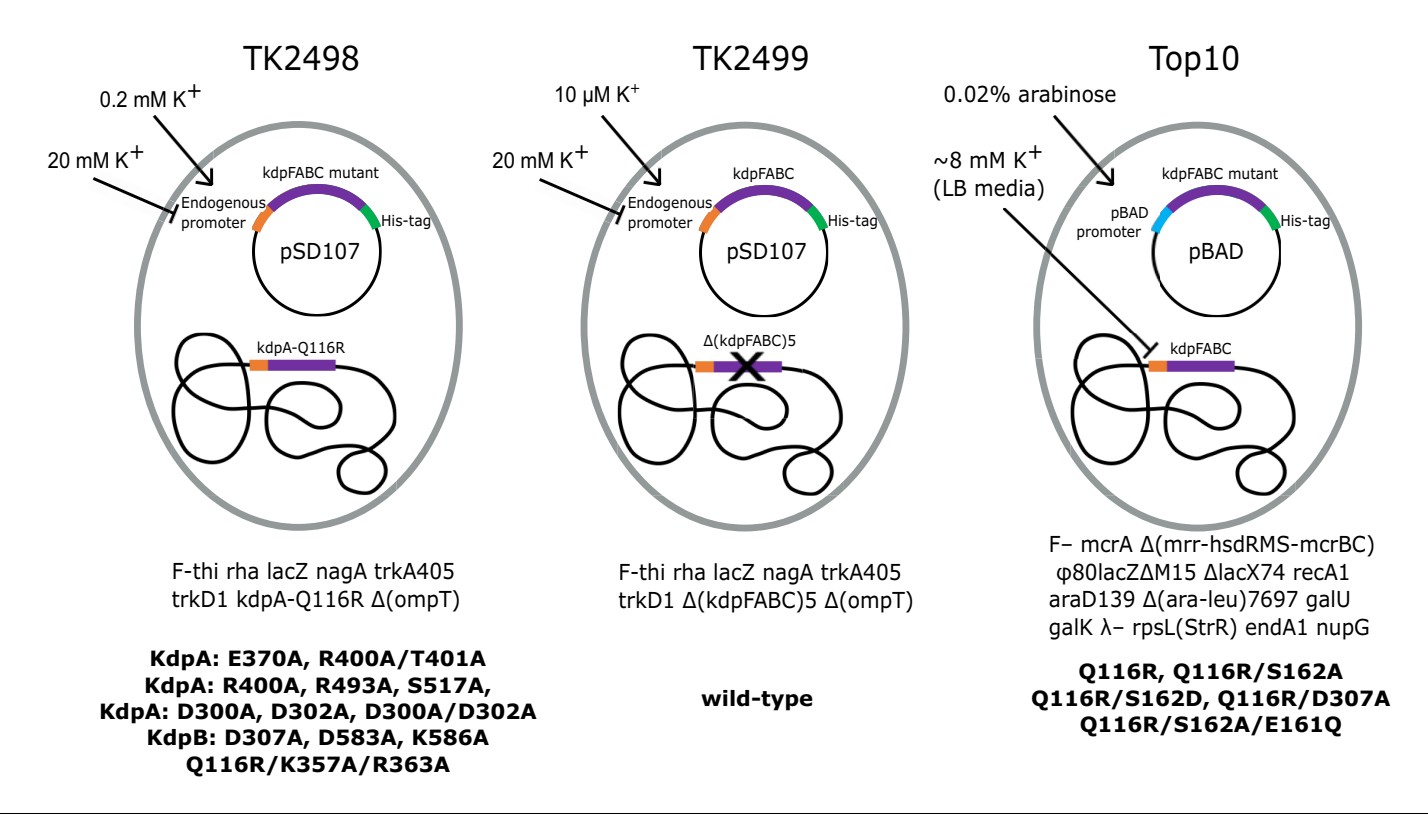

**Figure 2.** Bacterial strains used to express various KdpFABC mutants. The TK2498 strain contains a chromosomal copy of kdpFABC that carries the KdpA-Q116R mutation behind the native promotor. This gene rescues the cells in K$^+$-deficient media (0.2 mM), which also serves to induce expression not only of the chromosomal copy, but also of His-tagged mutants on plasmids derived from pSD107, which carries the same native promoter. The genotype is shown below together with mutants expressed from this strain. The TK2499 strain has the same genotype as TK2498, except that it lacks the chromosomal copy of kdpFABC. It was used for expression of the His-tagged WT protein, which is induced at μM K$^+$ concentrations. This strain was used for K$^+$ shock experiments. The Top10 strain was used for expression of several His-tagged mutants behind the pBAD promoter grown in standard LB media and induced with arabinose. This strain was used for expression of proteins for biochemical assays of aspartyl phosphate formation. The mutants and their expression conditions are summarized in *Table 1*.

strain. These results motivated us to examine whether phosphorylation of Ser162, and consequent inhibition of KdpFABC, was a cellular response to elevated K$^+$ in the media.

## Increased K$^+$ is a physiological stimulus for serine phosphorylation

To test whether elevated K$^+$ concentrations lead to serine phosphorylation, we first grew TK2499 cells with WT KdpFABC under K$^+$ limiting (micromolar) conditions and then split the culture into two batches immediately prior to harvest. To one batch, we added 20 mM K$^+$ - so called K$^+$ shock - and growth was continued for 20 min. KdpFABC was then purified from both batches. ATPase assays documented a 40% inhibition for protein derived from the cells subjected to the K$^+$ shock, which was fully restored after treatment with LPP (*Figure 4a*). Mass spectrometry confirmed that K$^+$ shock produced a higher level of serine phosphorylation: ion currents for the phosphorylated peptide relative to the unphosphorylated peptide were 27% for K$^+$ shock vs 1% for the control (*Figure 4b*). This quantification is nominal because phosphorylated and nonphosphorylated versions of the same peptide are not ionized and detected with equal efficiency and, as a result, positive ion mass spectrometry typically underestimates the level of phosphorylation (*Xu et al., 2005*). However, relative phosphorylation levels of different samples are reliable (*Huang et al., 2016*) and these data show that the K$^+$ shock resulted in much higher levels of Ser162 phosphorylation as well as inhibition of ATPase activity.

**Table 1.** KdpFABC mutants.

| Annotation | Mutations | Strain | Promoter | Condition | Figures |
|---|---|---|---|---|---|
| WT | - | TK2499 | KdpE | μM K$^+$ | 3,4,4-s1,6,6-s1 |
| KdpA-E370A | KdpA-E370A | TK2498 | KdpE | 0.2 mM K$^+$ | 3 |
| KdpA-R400A/T401A | KdpA-R400A KdpA-T401A | TK2498 | KdpE | 0.2 mM K$^+$ | 3 |
| KdpA-S517A | KdpA-S517A | TK2498 | KdpE | 0.2 mM K$^+$ | 3 |
| KdpA-R400A | KdpA-R400A | TK2498 | KdpE | 0.2 mM K$^+$ | 3 |
| KdpA-R493A | KdpA-R493A | TK2498 | KdpE | 0.2 mM K$^+$ | 3 |
| KdpB-D300A | KdpB-D300A | TK2498 | KdpE | 0.2 mM K$^+$ | 3 |
| KdpB-D302A | KdpB-D302A | TK2498 | KdpE | 0.2 mM K$^+$ | 3 |
| KdpB-D300A/D302A | KdpB-D300A/ KdpB-D302A | TK2498 | KdpE | 0.2 mM K$^+$ | 3 |
| KdpB-D307A | KdpB-D307A | TK2498 | KdpE | 0.2 mM K$^+$ | 3,6-s1 |
| KdpB-D583A | KdpB-D583A | TK2498 | KdpE | 0.2 mM K$^+$ | 3 |
| KdpB-K586A | KdpB-K586A | TK2498 | KdpE | 0.2 mM K$^+$ | 3 |
| Q116R/K357A/R363A | KdpA-Q116R/ KdpB-K357A/ KdpB-R363A | TK2498 | KdpE | 0.2 mM K$^+$ | 6-s2 |
| Q116R | KdpA-Q116R | Top10 | pBAD | LB media* | 5,5-s1,s2,6 |
| S162A/Q116R | KdpA-Q116R/ KdpB-S162A | Top10 | pBAD | LB media* | 5,5-s1,s2 |
| S162D/Q116R | KdpA-Q116R/ KdpB-S162D | Top10 | pBAD | LB media* | 5,5-s1 |
| D307A/Q116R | KdpA-Q116R/ KdpB-D307A | Top10 | pBAD | LB media* | 5,5-s1,s2 |
| S162A/E161Q/ Q116R | KdpA-Q116R/ KdpB-S162A/ KdpB-E161Q | Top10 | pBAD | LB media* | 5,5-s1,s2 |

*LB media has been reported to contain 8 mM K$^+$ (**Su et al., 2009**).

## Time dependence of serine phosphorylation

To extend these studies, we varied the length of time that cells were subjected to K$^+$ shock in order to characterize the time dependence of the response. For each experiment, the culture was split, 20 mM added to one half and incubated for either 0, 10, or 90 min before harvesting the cells. The incubation time is nominal given that cells had continued exposure to the media during the ensuing 20-min centrifugation step. ATPase inhibition increased as incubation time increased (15%, 25%, and 74%) and LPP treatment largely relieved this inhibition (*Figure 4c*). In addition to mass spectrometry, we used SDS-PAGE together with Phos-tag stain (*Figure 4—figure supplement 1*) to quantify levels of serine phosphorylation. Results from mass spectrometric phosphorylation analysis were consistent with this graded response to the K$^+$ shock (increased phosphorylation levels of 2.07-, 4.02-, and 5.23-fold, respectively) and confirmed Ser162 as the site of phosphorylation (*Figure 3—figure supplement 1*). However, data from Phos-tag staining was more consistent across independent experiments and these data clearly illustrate a time-dependent increase in serine phosphorylation and the inverse correlation with ATPase activity (*Figure 4d*).

In addition to these isolated experiments, we conducted a time course in which a single culture was split into four batches that were shocked for varying times. In addition to a control, untreated batch, the cultures were exposed to 20 mM K$^+$ for 0, 20 and 60 min prior to harvesting the cells. ATPase activities of the corresponding preparations of KdpFABC show an increasing amount of inhibition, which was largely reversed by treatment with LPP (*Figure 4e*), and increasing levels of serine phosphorylation (*Figure 4f*). Aggregation of data from all of these experiments showed a clear correlation between ATPase activity and serine phosphorylation for both mass spectrometry (*Figure 4g*) and Phos-tag staining (*Figure 4h*). Although the mass spectrometry data showed a lot

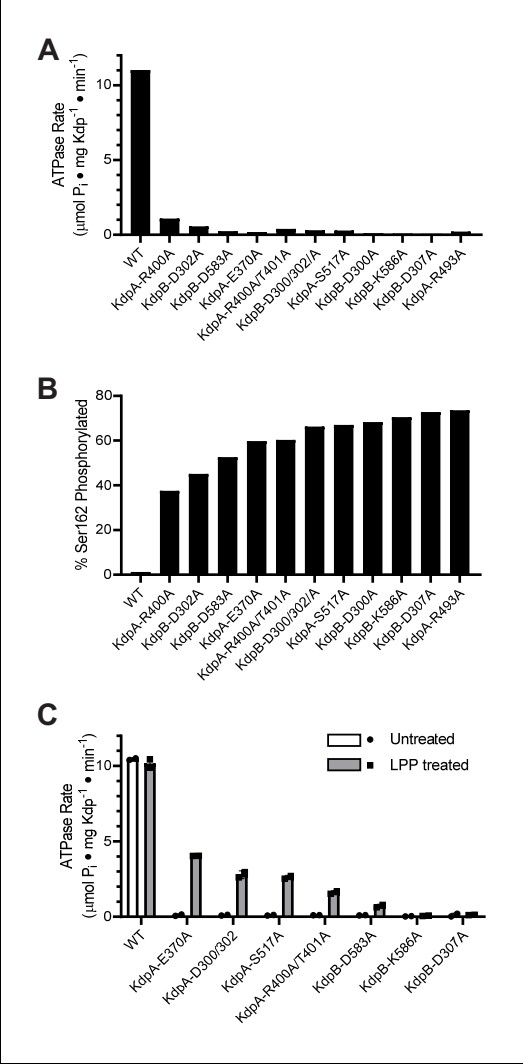

**Figure 3.** Survey of alanine substitution mutants reveals inhibition and serine phosphorylation of KdpB. (**A**) Very low levels of ATPase activity were obtained from a series of Ala substitution mutants. As indicated in *Table 1*, WT protein has been expressed in TK2499 at very low levels of K$^+$, whereas the other constructs were expressed in TK2498 at 200 μM K$^+$. (**B**) Mass spectrometry of these Ala mutants revealed high levels of phosphorylation of Ser162 on KdpB. (**C**) Treatment of several of these mutants with LPP to remove serine phosphorylation resulted in significant stimulation of ATPase activity, supporting the notion that serine phosphorylation is inhibitory. Data in panels A and B are single measurements, whereas data in panel C were collected in duplicate.

The online version of this article includes the following source data and figure supplement(s) for figure 3:

**Source data 1.** Survey of alanine substitution mutants.
**Figure supplement 1.** Detection of Ser162 phosphorylation by mass spectrometry.

of scatter, it was well correlated with measurements from Phos tag staining (*Figure 4i*). Combining the site specificity of mass spectrometry with the quantification accuracy of Phos-tag staining, provides strong evidence that a return to K$^+$-rich media causes phosphorylation of Ser162 with a concomitant and proportional inhibition of ATPase activity.

## Effect of serine phosphorylation on partial reactions

Transient formation of an aspartyl phosphate at the catalytic site in the P-domain of KdpB is a key step in the reaction cycle. To evaluate how serine phosphorylation affects individual steps in the cycle, we used a conventional assay employing [γ$^{32}$P]-ATP to initiate the cycle followed by acid quench at defined time intervals. This assay reports on steady-state levels of aspartyl phosphate (EP), which includes molecules in both E1~P or E2-P states (*Figure 1*). These two states were then distinguished by their reactivity to ADP, based on the fact that the high-energy form (E1~P) is readily dephosphorylated by ADP, whereas the low-energy form (E2-P) is not (*Toustrup-Jensen et al., 2001*). For these studies, we introduced a number of mutations (*Table 1*) to elucidate the effect of serine phosphorylation, a stable post-translational modification that is distinct from the transient phosphorylation of the catalytic Asp307. In particular, the KdpB-S162A mutation prevented serine phosphorylation and the KdpB-S162D mutation served as a surrogate for the fully phosphorylated protein. The KdpB-E161Q mutation is expected to block hydrolysis of the E2-P species (step four in *Figure 1*), based on work with other P-type ATPases (*Clausen et al., 2004*), and KdpB-D307A serves as a negative control that prevents EP formation altogether. All these mutations in KdpB were combined with the KdpA-Q116R mutation, denoted kdp-42 or strain TK2242-42 in previous work (*Epstein et al., 1978*; *Siebers and Altendorf, 1989*), which lowers K$^+$ affinity to the millimolar range and facilitates study of K$^+$-dependence by eliminating the background activity generally observed with the WT construct (*Siebers and Altendorf, 1988*; *Siebers and Altendorf, 1989*). Although it may be surprising that a substitution in the signature TGES sequence is tolerated, an extensive phylogenetic analysis of the P-type ATPase superfamily indicates that the serine has the greatest variability in this motif and appears as Thr, Ala, Arg, or Pro in other family members (*Chan et al., 2010*). All relevant constructs were expressed behind a

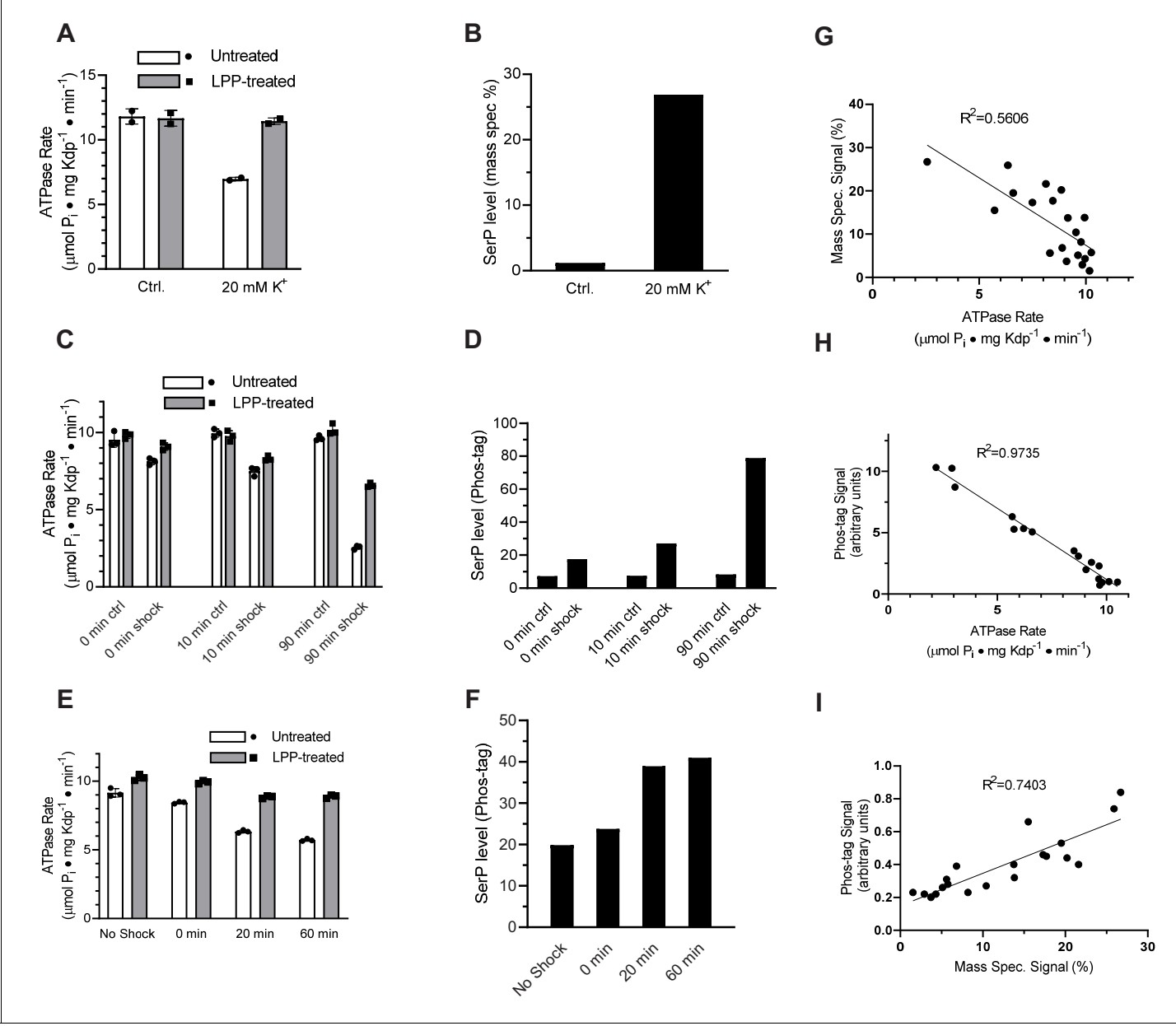

**Figure 4.** K⁺ shock leads to serine phosphorylation of KdpB and inhibition of ATPase activity. For these experiments, TK2499 cells were cultured at low K⁺ concentrations to induce expression of WT kdpFABC; thereafter, the culture was split and one half was subjected to 20 mM K⁺ for varying periods of time. (**A**) A 20 min K⁺ shock resulted in loss of ~40% of ATPase activity, which was fully recovered by LPP treatment. (**B**) Serine phosphorylation levels for samples from panel A determined by mass spectrometry were much higher after K⁺ shock. (**C**) ATPase rates from additional individual experiments with varying periods of K⁺ shock. The time periods are nominal because harvest of cells required a 20-min centrifugation in their respective media. (**D**) Serine phosphorylation levels for samples in panel C determined by Phos-tag staining of SDS gels show an increase in serine phosphorylation levels that is proportional to the observed inhibition of ATPase activity. (**E**) ATPase activities from a single culture split into four batches and treated with K⁺ for variable times show a graded inhibitory response. (**F**) Levels of serine phosphorylation for the cultures in panel E were determined by Phos-tag staining. (**G**) Data from all experiments were combined and show a reasonable correlation (R = 0.56) between ATPase activity and levels of serine phosphorylation determined by mass spectrometry. (**H**) Correlation of ATPase activities from all the experiments with levels of serine phosphorylation determined by Phos-tag staining produces a much better correlation (R = 0.97). (**I**) Correlation of serine phosphorylation levels from mass spectrometry and Phos-tag staining. ATPase measurements were performed in triplicate and shown as mean plus SEM; phosphoserine levels represent single measurements.

The online version of this article includes the following source data and figure supplement(s) for figure 4:

**Source data 1.** K⁺ shock leads to serine phosphorylation of KdpB and inhibition of ATPase activity.

**Figure supplement 1.** Preparation of KdpFABC and analysis of serine phosphorylation using Phos-tag stain.

pBAD promoter in K⁺ rich media (*Figure 2*); these conditions activated the pathway for serine phosphorylation, which is evident from the stimulation in ATPase activities that was observed after treatment of the KdpA-Q116R construct with LPP (*Figure 5—figure supplement 1*). No inhibition was seen when Ser162 was mutated to Ala (KdpB-S162A) and mutations of KdpB-D307A, KdpB-E161Q and KdpB-S162D all resulted in complete inhibition of ATPase activity even after LPP treatment.

Typical experiments are shown in *Figure 5a and b*, which represent the time course of EP formation at room temperature in the absence and presence of 10 mM K⁺, respectively. In the presence of K⁺, EP levels of active constructs (e.g. S162A/Q116R) decay over time as the ATP in the solution is exhausted (*Figure 5b*). In contrast, EP levels were stable in the absence of K⁺, because there is no enzymatic turnover (*Figure 5a*), and at 4°C, because of a much slower turnover rate (*Figure 5—figure supplement 2*). EP levels were also stable for inactive constructs such as S162D/Q116R and E161Q/S162A/Q116R. Data from 15 individual experiments have been normalized and pooled in *Figure 5c*. These data included all points for experiments with low turnover rates (i.e. at 4°C or lacking K⁺) and initial points for experiments with high turnover rates (i.e. at room temperature in the

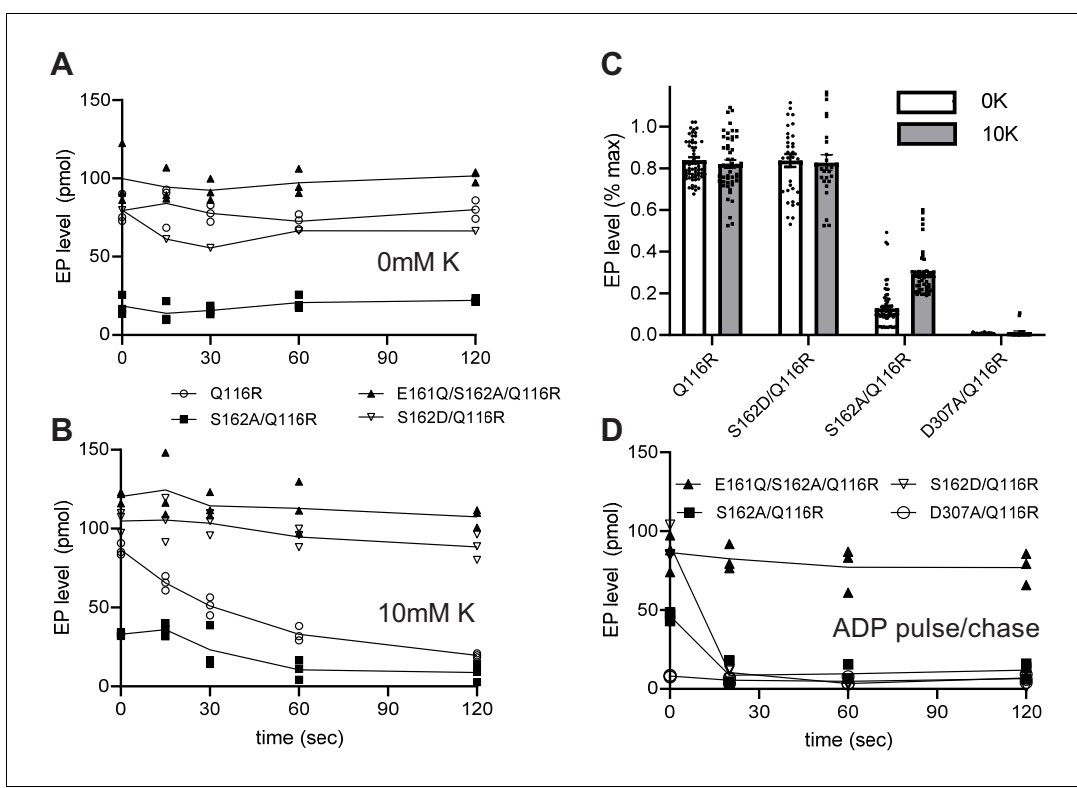

**Figure 5.** Steady-state levels of aspartyl phosphate (EP). (**A–B**) Individual experiments comparing time-dependence of EP formation at room temperature of several different constructs in the absence (**A**) or presence (**B**) of 10 mM K⁺. EP formation was initiated by addition of [γ³²-P]-ATP and aliquots were taken at various time intervals and quenched with TCA. The constructs are indicated in the legend. (**C**) Aggregated data from 15 individual experiments that have been normalized relative to levels for the E161Q construct. (**D**) Pulse-chase experiment in which samples have been incubated for 15 s with [γ³²-P]-ATP at 4°C to achieve maximal levels of EP (plotted at time zero) followed by addition of 10 mM ADP and 10 mM EDTA for the indicated time periods. Data was collected in triplicate and lines are drawn through mean values at each time point. Error bars in panel C correspond to SEM.

The online version of this article includes the following source data and figure supplement(s) for figure 5:

**Source data 1.** Steady-state levels of aspartyl phosphate (EP).
**Figure supplement 1.** Preparations used for EP measurements.
**Figure supplement 1—source data 1.** Preparations used for EP measurements.
**Figure supplement 2.** Steady-state levels of aspartyl phosphate (EP) are constant at 4°C.
**Figure supplement 2—source data 1.** Steady-state levels of aspartyl phosphate (EP) at 4°C.

presence of $K^+$) or pulse chase (see below). Normalization of data from each experiment was based on data from E161Q/S162A/Q116R, which consistently produced the highest EP levels and which were stable under all the experimental conditions that were tested. Thus, data in *Figure 5c* indicate that Q116R and S162D constructs, which both reflect the effects of Ser162 phosphorylation, produced high levels of EP both in the presence and absence of $K^+$. In contrast, EP levels for the Ser162A construct, which cannot be serine phosphorylated, are much lower and showed a significant dependence on $K^+$. As expected, the D307A construct produced negligible levels of EP. These data suggest that formation of the high-energy aspartyl phosphate (step two in *Figure 1*) is $K^+$-dependent and that phosphorylation of Ser162 not only uncouples this step but also prevents the hydrolysis of the phosphoenzyme intermediate.

The data from S162D/Q116R and E161Q/S162A/Q116R suggest that the KdpB-S162D mutation and the KdpB-E161Q mutation cause the cycle to be blocked at some point after aspartyl phosphate formation. In order to determine more precisely where the cycle was blocked, we used a pulse-chase strategy to distinguish E1~P from E2-P intermediates. For this assay, we initiated phosphoenzyme formation in the presence of $K^+$ with $[\gamma^{32}P]$-ATP at 4°C where steady-state levels are stable. After 15 s, an aliquot was removed (plotted as time zero in *Figure 5d*) and a solution containing 10 mM ADP and 10 mM EDTA was added to determine the proportion of E1~P present. *Figure 5d* shows that EP levels for S162A/Q116R and S162D/Q116R are immediately reduced to zero (within the time resolution of this assay). This result indicates that E1~P is the prevalent species at steady state and that serine phosphorylation prevents the molecule from progressing to the E2-P state (step three in *Figure 1*). In contrast, the E161Q/S162A/Q116R construct was completely unreactive with ADP, indicating the the E2-P state was trapped in accordance with the effects of the Glu to Gln mutation on other P-type pumps (*Clausen et al., 2004*).

Although steady levels of EP from S162D/Q116R are consistent with blockage of the E1~P to E2-P transition, data from the Q116R construct reflect a more complex situation. Unlike the S162D/Q116R construct, the Q116R construct generates a mixture of active (no serine phosphorylation) and inactive (serine phosphorylated) molecules. At first glance, one might expect that the fraction of inactive molecules with serine phosphorylation would produce stable EP levels while unmodified, active molecules would turnover and very slowly exhaust the ATP in the solution. However, ADP is produced as the active molecules turnover which will then react with and dephosphorylate the molecules arrested in the E1~P state, as seen in the pulse chase experiments. This behavior would then account for the observed fall in overall EP levels of partially phosphorylated Q116R mutants over time.

## Effect of serine phosphorylation on ligand binding and transport

An alternative explanation for the observed inhibition could be that serine phosphorylation simply slows down one of the steps of the reaction cycle or changes the affinity for one of the substrates. To evaluate this alternative, we used ATPase activity to estimate affinities for $K^+$ and ATP as well as inhibition by orthovanadate. Orthovanadate acts as a transition state analogue for inorganic phosphate and typically binds to the E2 state of P-type ATPases to form an inhibitory complex (*Clausen et al., 2016*). In this way, the $IC_{50}$ for inhibition by orthovanadate reflects not only an intrinsic binding affinity but also the conformational equilibrium between E2 and E1 states of P-type ATPases (*Toustrup-Jensen et al., 2001*). For the $K^+$ titration, we used the KdpA-Q116R mutant and for the ATP and vanadate titrations we used WT protein, both with high levels of serine phosphorylation, evident from a threefold to fivefold stimulation after LPP treatment (e.g. $V_{max}$ = 1.9 and 9.3 μmoles/mg/min in *Figure 6b*). Fitting of data from untreated and treated samples (*Figure 6a & b*) indicated that differences in $K_M$ for $K^+$ (21 mM and 7 mM, respectively) and ATP (60 μM and 72 μM) were not statistically significant. Similarly, inhibition by orthovanadate under standard ATPase conditions (150 mM $K^+$ and 2.4 mM ATP) indicate no significant change in $IC_{50}$ (9.2 μM vs. 8.4 μM). Although not a comprehensive analysis of reaction cycle kinetics, these results reveal no significant changes in the affinity of these substrates and are therefore consistent with the interpretation that the KdpA-Q116R preparation consists of a fraction of molecules with normal binding affinities and a fraction of inactive molecules with serine phosphorylation. Although we consider it unlikely, it is possible that the Q116R mutation is masking an effect of serine phosphorylation on $K^+$ affinity of the WT protein, which is expected to be in the micromolar range (*Buurman et al., 1995*; *Dorus et al., 2001*).

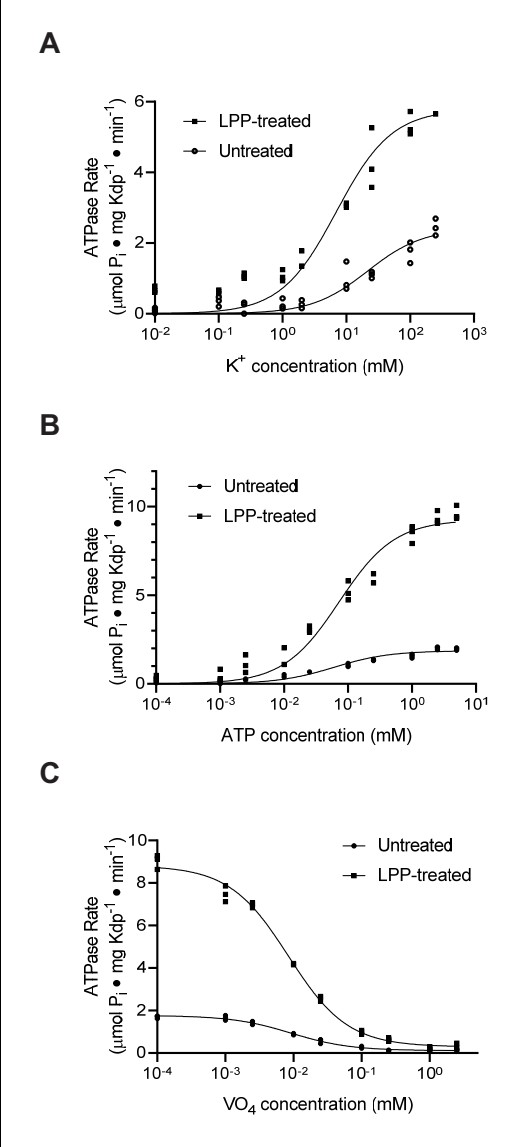

**Figure 6.** Substrate affinities are not affected by serine phosphorylation. (**A**) For K$^+$ titrations, the KdpA-Q116R mutant was expressed with the pBAD system, which produced high levels of serine phosphorylation (see **Figure 5—figure supplement 1**). K$^+$-dependence curves show that TPP treatment increased V$_{max}$ significantly from 2.4 to 5.8 μmoles/mg/min, but that differences in K$_M$ (21.5 mM vs. 7.0 mM) where not statistically significant at the 95% confidence level. (**B**) ATP dependence of WT protein was fitted with V$_{max}$ = 1.9 and 9.3 μmoles/mg/min for untreated and treated samples respectively with insignificant change in K$_M$ (60 vs 73 μM). (**C**) Titrations of WT protein with orthovanadate produced maximal activities of 1.7 and 8.7 μmoles/mg/min for untreated and treated samples and IC$_{50}$ values of 9.2 and 8.4 μM, respectively. Data were collected in triplicate and fitted with the Michaelis Menton equation (**A,B**) and inhibitor-response equation (**C**) in Prism8.

*Figure 6 continued on next page*

Membrane transport assays indicate that transfer of K$^+$ across the membrane remain coupled to ATPase activity of the complex. For these assays, KdpFABC was reconstituted into proteoliposomes and a voltage-sensitive dye (DiSC3) was used to report electrogenic transport of K$^+$ into the vesicles (*Fendler et al., 1996*). Upon addition of ATP, a robust decrease in fluorescence indicated a buildup of a positive membrane potential (*Figure 6—figure supplement 1*). The initial part of the curve was fitted with an exponential to quantify the rate (*Damnjanovic and Apell, 2014a*) and to compare the preparation before and after LPP treatment. This analysis shows an excellent correlation between ATPase activity and transport. If an uncoupling of serine-phosphorylated molecules allowed passive transport of K$^+$ in the absence of ATPase activity, the membrane voltage generated by unphosphorylated molecules would be rapidly dissipated, as is seen by the addition of valinomycin in *Figure 6—figure supplement 1a* (red arrow). Thus, these data suggest that the fraction of active molecules are fully capable of coupled transport whereas the fraction of inactive molecules with serine phosphorylation do not allow passive transport of K$^+$ across the membrane.

Finally, we tested our previous hypothesis for inhibition, involving salt bridges between the phosphorylated Ser162 in the A-domain and basic resides in the N-domain. Specifically, Lys357 and Arg363 were mutated to Ala to prevent this interaction. The corresponding construct, Q116R/K357A/R363A was expressed under K$^+$-replete conditions, resulting in substantial levels of serine phosphorylation as shown by Phos-Tag staining. (*Figure 6—figure supplement 2*). Comparison of ATPase activity before and after LPP treatment document a robust inhibition by serine phosphorylation despite the inability to form these salt bridges, indicating that a different inhibitory mechanism is in play.

## Discussion

In this study, our goal was to explore the effects of serine phosphorylation on KdpFABC that was first described by *Huang et al., 2017*. In a physiological context, it is well established that kdpFABC expression is induced by K$^+$-deficient media in order to maintain sufficient levels of K$^+$ inside the cell. Our data show that phosphorylation of Ser162 on KdpB occurs when these cells are returned to K$^+$-rich media. This posttranslational modification results in inhibition of ATPase activity as well as K$^+$ transport, thus preventing

*Figure 6 continued*

The online version of this article includes the following source data and figure supplement(s) for figure 6:

**Source data 1.** Substrate affinities are not affected by serine phosphorylation.
**Figure supplement 1.** Electrogenic transport of K⁺ by KdpFABC.
**Figure supplement 1—source data 1.** Electrogenic transport of K⁺by KdpFABC.
**Figure supplement 2.** Interactions between A- and N-domains are not responsible for the inhibitory effects of serine phosphorylation.
**Figure supplement 2—source data 1.** Mutation of K357 and R363 in KdpB has no effect on the inhibitory effects of serine phosphorylation.

unnecessary ATP consumption and uncontrolled increases in intracellular $K^+$. This novel inhibitory mechanism explains earlier studies in which cell-based $K^+$ uptake was inactivated under similar conditions (*Rhoads et al., 1978*; *Roe et al., 2000*). The work by *Roe et al., 2000* also showed that inactivation was irreversible in situ, that the effect was specific to $K^+$ and was not due to osmotic upshock, which in any case would be very modest with the addition of only 20 mM KCl. Although these authors showed complete inhibition in cell-based assays of transport, we achieved only ~75% maximal inhibition of purified protein after much longer incubation times in $K^+$-rich media (90 min vs. 2 min). This difference might reflect the much higher expression levels used for the current study. Specifically, we employed a multicopy plasmid in order to produce high levels of protein expression suitable for biochemical purification compared to the single chromosomal copy used in previous work. The higher expression level might affect the extent of the phosphorylation either by exceeding the capacity of the (still unknown) kinase system or by changing the ionic balance of cells and thus influencing the sensing mechanism. In any case, the inability to completely inhibit KdpFABC is likely to be detrimental to the cells, either by exhausting ATP levels or by creating an ionic imbalance. Either effect could interfere with progressive inhibition during long $K^+$ shock experiments (e.g. the 90 min time point in *Figure 4c*) and affect the overall viability of the culture.

P-type ATPases are subject to a variety of regulatory strategies, but the inhibition of KdpFABC by phosphorylation of the highly conserved TGES motif represents a novel approach for this superfamily. In the case of plasma-membrane $Ca^{2+}$-ATPase (PMCA) (*Calì et al., 2017*), yeast and plant $H^+$-ATPases (*Haruta et al., 2015*), and the yeast lipid flippase (Drs2p) (*Jacquot et al., 2012*), the pumps are held in autoinhibited states by C-terminal domains. Similarly, phospholamban (PLB) and sarcolipin (SLN) are regulatory subunits that inhibit $Ca^{2+}$-ATPase from sarcoplasmic reticulum (SERCA) (*Primeau et al., 2018*) and FxYD proteins regulate Na/K-ATPase (*Garty and Karlish, 2006*). In most cases, these regulatory domains/subunits interact with the conserved catalytic domains to prevent turnover; distinct physiological factors bind to or modify these regulatory domains/subunits to disrupt these interactions and allow catalytic domains to function. These stimulatory factors are diverse and include $Ca^{2+}$/calmodulin for PMCA, phosphatidylinositol-4-phosphate for Drs2p and serine phosphorylation for the others. The regulatory mechanism for KdpFABC is novel in that the modification inhibits rather than stimulates the transport cycle and that the modification is applied directly to one of the catalytic domains rather than to an accessory structural element.

Although serine phosphorylation is well known in eukaryotes, there are relatively few examples in bacteria. More commonly, signaling cascades in bacteria involve phosphorylation of histidine or aspartate residues, typically involving two-component systems such as KdpD/E. Correspondingly, few Ser/Thr kinases have been identified in bacteria. In fact, a catalogue of bacterial signal transduction systems lists only two eukaryotic-like Ser/Thr kinases in *E. coli*: ubiB and yegI (*Galperin et al., 2010*). The former is involved in the ubiquinone biosynthesis pathway (*Poon et al., 2000*), whereas the latter has been characterized in vitro but its function in vivo remains unknown (*Rajagopalan et al., 2018*). These Ser/Thr kinases have been proposed to originate in eukaryotes and their presence in bacteria to be the result of horizontal gene transfer (*Lai et al., 2016*). If so, it would seem unlikely that a process as fundamental as osmotic homeostasis would be acquired by this mechanism. Atypical Ser/Thr kinases have also been reported in prokaryotes, such as the Hpr kinase that controls metabolism of carbon sources in gram positive bacteria, SpoIIAB, RsbT and RsbW that act on sigma factors to control sporulation and stress response in *B. subtilis* (*Pereira et al., 2011*), and AceK in *E. coli* that phosphorylates isocitrate dehydrogenase to control metabolism through the TCA cycle (*Cortay et al., 1988*). Their involvement in regulation of KdpFABC is worth considering. An intriguing alternative is that KdpB engages in

autophosphorylation; if so, it must be very inefficient or require special conditions, because self-inhibition did not occur over the time course of our ATPase assays, nor has it been reported in the literature. High levels of phosphorylation observed on the D307A/Q116R mutant (*Figure 3b*) preclude transfer of phosphate from Asp307 to Ser162. The alternative is that phosphate transfers directly from ATP to Ser162, but the complete lack of signal in EP formation experiments using the D307A mutation (*Figure 5*) is inconsistent with this idea. Another intriguing possibility is that KdpD, which is already designed to sense the $K^+$ need of the cell (*Schramke et al., 2016*), would regulate activity of a Ser/Thr kinase that targets KdpFABC, as has been seen in several other systems (*Pereira et al., 2011*).

The dependence of serine phosphorylation on cell growth conditions and its inhibitory effects on KdpFABC has important ramifications for our understanding of the reaction cycle. Earlier biochemical studies concluded that formation of E1~P was $K^+$-independent, that $K^+$ bound to the E2-P state and was transported across the membrane during hydrolysis of the aspartyl phosphate (*Damnjanovic and Apell, 2014b*; *Siebers and Altendorf, 1989*). According to this scheme, $K^+$ transport by KdpFABC is analogous to $K^+$ transport by $Na^+/K^+$-ATPase (*Goldshleger et al., 2001*) or lipid by lipid flippases (*Jacquot et al., 2012*). However, this conclusion is at odds with recent structural studies in which KdpB adopted an E1-like structural state when $K^+$ was bound in the selectivity filter of KdpA (*Huang et al., 2017*; *Stock et al., 2018*). In the current biochemical experiments, we also observed $K^+$-independence of EP formation, but only when KdpB was serine phosphorylated or carried the S162D mutation to mimic serine phosphorylation. In contrast, when the S162A mutation was used to prevent this phosphorylation, we observed much lower levels of EP formation and a clear dependence on the presence of $K^+$. The lower levels of EP for the S162A construct in the presence of $K^+$ likely reflects catalytic turnover of these molecules which reduces the fraction of time spent in one of the phosphoenzyme states (E1~P or E2-P). In contrast, inactive constructs such as S162D, E161Q, and the serine phosphorylated species are essentially stuck in one of these phosphoenzyme states. ADP-sensitivity seen in the pulse chase experiments show that E1~P is the prevalent species and that serine phosphorylation prevents, or at least greatly slows down, the E1~~P to E2-P transition. In contrast, the E161Q mutation typically interferes with hydrolysis of the aspartyl phosphate (*Clausen et al., 2004* #2555), thus preventing the E2-P to E2 transition and, as expected, the phosphoenzyme formed by this construct does not react with ADP. Therefore, we conclude that formation of E1~P by functional KdpFABC molecules is dependent on $K^+$ binding by KdpA from the periplasm and that $K^+$ is likely imported across the membrane during the transition from E1~P to the E2-P state.

In addition to this inhibition of catalytic activity, serine phosphorylation of KdpFABC also eliminates the $K^+$-dependence of EP formation. As described above, the inhibited pump is quite capable of reacting with ATP to form aspartyl phosphate intermediates, but has lost the $K^+$ dependence of this step. Interestingly, the E161Q and S162D mutations also resulted in uncoupling, which suggests that this crucial TGES[162] loop has long-range allosteric effects over $K^+$ sensing within the transmembrane domain. Although the path taken by $K^+$ across the membrane is still uncertain (*Pedersen et al., 2019*), the canonical substrate-binding site on M4 is connected directly to the P-domain and is likely to play an important role in sensing $K^+$ and initiating EP formation. In addition, work on other P-type ATPases has shown how large movements of the A-domain propagate to the transmembrane domain via its linkages to M1 and M2 helices. These helices control occlusion of $Ca^{2+}$ in the M4 site by SERCA (*Sørensen et al., 2004*) and access of phosphatidyl serine to this site by lipid flippases (*Hiraizumi et al., 2019*). The unusual pose of the A-domain in the crystal structure of serine-phosphorylated KdpFABC indicates that it has a larger range of motion relative to these other pumps. Thus, the loss of $K^+$ dependence by serine phosphorylated KdpFABC could reflect enhanced mobility of the A-domain and associated structural elements that allow cytosolic ions access to the canonical binding site, thereby triggering the conformational change leading to E1~P in an unnatural and non-selective way. Although not observed in other P-type ATPases, the KdpB-E161Q mutation appears to have an analogous effect on the $K^+$-dependence of this first step. Once E1~P is formed, our biochemical data suggest that serine phosphorylation prevents KdpFABC from progressing to the E2-P state. E2-P is characterized by the loss of ADP sensitivity, which results from an interaction between the TGES[162] loop and the aspartyl phosphate (Asp307). It seems plausible that a phosphate moiety on Ser162 would sterically interfere with this interaction, thus preventing formation of E2-P. For the Q116R/S162A/E161Q, the lack of serine phosphorylation allows this

construct to achieve the E2-P state, but the isosteric E161Q mutation prevents hydrolysis of the aspartyl phosphate that is normally orchestrated by this residue. Finally, although serine phosphorylation eliminates the $K^+$-dependence of EP formation, the complex is not truly uncoupled since transport of $K^+$ across the membrane, both active and passive, is also inhibited.

This mechanism of inhibition differs from our earlier speculation that salt bridges between the serine phosphate and the N-domain simply held the protein in the inactive conformation seen in the X-ray structure (*Huang et al., 2017*). Indeed, this interaction was not seen in the more recent cryo-EM structures (*Stock et al., 2018*) and we found that mutations designed to break up this interaction (K357A/R363A) had no effect on the inhibitory properties of serine phosphorylation. This finding is not surprising given the lack of conservation in this region of the N-domain and specifically for the two basic residues (K357 and R363) that mediated the interaction in the crystal structure. Interestingly, the cryo-EM structures show that both E1 and E2-P like conformations can be adopted by serine phosphorylated KdpFABC, which might appear to contradict our biochemical data. However, our studies of EP formation were designed to study the cycle initiated by ATP addition (clockwise in *Figure 1*), whereas E2-P can also be formed directly from phosphate by so-called back-door phosphorylation (reversal of steps 4 and 5 in *Figure 1*). Although this step is energetically unfavorable, it may have been facilitated by the presence of $AlF_4$ in samples used for the cryo-EM studies.

In conclusion, our work has characterized a novel mechanism for regulating P-type ATPases and maintaining $K^+$ homeostasis in bacteria. Fundamental questions remain regarding the molecular mechanism of KdpFABC, including the role of a unique tunnel that runs between the subunits and the precise path taken by $K^+$ as it crosses the membrane. Given the effects that we have documented on the enzymatic cycle, efforts should be made to ensure that serine phosphorylation is not a factor in future structural and biochemical work on this intriguing membrane pump.

## Materials and methods

### Expression of KdpFABC complex

For studies of serine phosphorylation, the pSD107 plasmid was transformed into *E. coli* strain TK2499 (both from W. Epstein, Univ. of Chicago), which lacks all other $K^+$ transport systems with genotype F⁻, thi, rha, lacZ, nagA, trkA405, trkD1, Δ(kdpFABC)5, Δ(ompT) (*Figure 2*). This plasmid was originally derived from pBR322 and contained the kdpFABC operon along with its endogenous promoter and an 8x histidine tag at the C-terminus of kdpC. Cells were grown in 4 L flasks at 31°C in K0-medium (46 mM $Na_2PO_4$, 23 mM $NaH_2PO_4$, 25 mM $(NH_4)_2SO_4$, 0.4 mM $MgSO_4$, 6 μM $FeSO_4$, 1 mM sodium citrate, 0.2% glucose, 1 μg/mL thiamine, 50 μg/mL carbenicillin) supplemented with additions of 10 μM KCl every 90 min after culture density reached an $OD_{600}$ of ~0.1. Total culture volumes ranged between 4 and 9 L. Cells were harvested at an $OD_{600}$ of ~0.3–0.4 by centrifugation at 3500 g for 20 min; the resulting cell pellets were frozen at −80°C for storage. For $K^+$ shock experiments, the culture was divided into equal portions and 20 mM KCl was added from a 2M stock solution followed by further incubation for defined periods of time prior to harvest.

To generate site-specific Ala mutants, site-directed PCR mutagenesis was used to alter codons in the WT operon on the plasmid described above. These plasmids were transformed into *E. coli* strain TK2498 with genotype F⁻, thi, rha, lacZ, nagA, trkA405, trkD1, kdpA-Q116R, Δ(ompT), also from W. Epstein (*Figure 2*). Unlike TK2499, this strain carries a chromosomal copy of kdpFABC with the Q116R mutation on kdpA, which lowers potassium affinity to ~6 mM and has been identified as kdp-42 or kdpA42 in previous publications (*Epstein et al., 1978*; *Siebers and Altendorf, 1989*). This tag-less copy served to ensure cell survival in the low $K^+$ conditions required to induce pump expression in cases where the alanine mutant complexes were inactive. For expression, cells were incubated overnight at 37°C in 10 mL K5-medium (K0-medium supplemented with 5 mM KCl). This culture was transferred to 500 mL K1-medium (K0-medium supplemented with 1 mM KCl) and incubated at 37°C for 8 hr. The 500 mL cell culture was transferred again to 18 L K0.2-medium (K0-medium supplemented with 0.2 mM KCl) and incubated at 31°C to induce expression of kdpFABC. Cells were harvested at a culture density of $OD_{600}$ = ~0.8 by centrifugation at 3500 g for 20 min at 4°C and cell pellets were either frozen or used immediately for purification.

For studies of EP formation, the kdpFABC operon with the 8x histidine tag was cloned into the pBAD vector (Invitrogen, Carlsbad, CA) and introduced into *E. coli* strain Top10 (Invitrogen). Cells

were grown in 4L flasks at 37°C in LB media supplemented with 100 mg/mL ampicillin to an $OD_{600}$ of 0.5. Expression was then induced by addition of 0.02% arabinose and the culture continued to grow at 37°C for 4 hr. Cells were then harvested by centrifugation as above and frozen for storage. Total culture volumes were typically 2 L.

## Purification of KdpFABC

For purification of the KdpFABC complex, cells were resuspended in 50 mM Tris pH 7.5, 1.2 M NaCl, 10 mM $MgCl_2$, 10% glycerol, protease inhibitor tablets (Roche, Basel Switzerland) and 1 mM DTT, and lysed with an Emulsiflex C3 high-pressure homogenizer (Avestin, Ottawa Canada). Whole cells and debris were removed by centrifugation for 15 min at 12,000 g, and membranes were pelleted by centrifugation for 2.5 hr at 90,140 g. Membranes were solubilized by overnight incubation in 50 mM Tris pH 7.5, 600 mM NaCl, 10 mM $MgCl_2$, 10% glycerol, 1 mM TCEP and 1.2% n-decyl-β-maltoside (DM) at 4°C using 20 mL per gram of membrane. After centrifugation for 30 min at 90,140 g, the solution was loaded on a 5 mL Ni-NTA HiTrap chelating column (GE Healthcare, Chicago, IL) that was equilibrated in Ni-NTA base buffer (50 mM Tris pH 7.5, 600 mM NaCl, 10 mM $MgCl_2$, 10% glycerol, 1 mM TCEP, and 0.15% DM) supplemented with 20 mM imidazole. The column was washed with 20 mL of this same buffer, followed by 20 mL of Ni-NTA base buffer supplemented with 50 mM imidazole. Stepwise 5 mL elutions (collected in 1.5 mL fractions) were then performed as the imidazole concentration was incremented in steps of 50 mM up to a final concentration of 300 mM imidazole. Alternatively, elution was done with a continuous gradient running from 50 to 500 mM imidazole. The fractions containing KdpFABC were pooled, concentrated to ~0.7 mL using a 100 kDa cut-off Amicon centrifugal filter (Sigma Aldrich, St. Louis, MO) and introduced onto a Superdex 200 size exclusion column (GE Healthcare) pre-equilibrated with 25 mM Tris pH 7.5, 100 mM KCl, 10% glycerol, 1 mM TCEP, and 0.15% DM. For measurements of EP formation, the NaCl was substituted for KCl during this purification. Fractions were either stored for short periods (days) at 4°C or frozen in liquid nitrogen and stored at −80°C.

## Quantification of serine phosphorylation

The Phospho-Tag phosphoprotein gel stain kit (ABP Biosciences, Beltsville, MD) was used to evaluate levels of serine phosphorylation. Specifically, 20 μg of precipitated protein was resuspended in sodium dodecylsulfate (SDS) buffer and run on a 10% SDS-PAGE gel. After staining according to manufacturer's instructions, gels were imaged in a Chemidoc imaging system (BioRad, Hercules, CA) using the epi-green illumination for Cy3 fluorescence. KdpB-S162A was used as a negative control and as background in quantifying band intensities.

Localization of the site of serine phosphorylation and estimation of stoichiometry were done by nano-liquid chromatography-tandem mass spectrometry (nano-LC-MS/MS). Slices from SDS gels were digested with 500 ng trypsin solution (10 ng/μL) (Promega, Madison, WI) in 50 mM $NH_4HCO_3$ and incubated for 4 hr at 37°C, followed by 500 ng GluC or AspN protease (Promega) overnight. Digestion was stopped with 10% formic acid and peptides extracted with 5% formic acid in 50% acetonitrile, followed by final extraction with acetonitrile. Combined peptide pools were concentrated to a small droplet by centrifugation under vacuum. The resulting peptides were desalted using a Stage Tip manually packed with Empora C18 High-Performance Extraction Disks (3M) (*Rappsilber et al., 2007*). Desalted peptide pools were dried under vacuum to a small droplet, and finally reconstituted to 20 μL with 0.1% formic acid.

LC-MS/MS was performed with a ThermoEasy nLC 1000 ultra-high-pressure UPLC system (ThermoFisher Scientific, Waltham MA) coupled online to a Q Exactive HF mass spectrometer with Nano-Flex source (ThermoFisher Scientific). Analytical columns (25–30 cm long and 75 μm inner diameter with 8 μm spray needle) were packed in-house with ReproSil-Pur C18 AQ 3 μm reversed-phase resin (Dr. Maisch GmbH, Ammerbuch-Entringen, Germany). The analytical column was placed in a column heater (Sonation GmbH, Biberach, Germany) regulated to a temperature of 45°C. The peptide pool was loaded onto the analytical column with buffer A (0.1% formic acid) at a maximum back-pressure of 300 bar; peptides were eluted with a 2-step gradient of 3% to 40% buffer B (100% ACN and 0.1% formic acid) over 40 min and 40% to 90% B over 5 min, at a flow rate of 250 nL/min over 60 min before direct infusion into the mass spectrometer. Mass spectrometric data were acquired using a data-dependent (DDA) top-15 method, dynamically choosing the most abundant, not-yet-

sequenced precursor ions from the survey scans (300–1750 Th). Peptide fragmentation was performed via higher energy collisional dissociation with a target value of $10^6$ ions. Precursors were isolated with a window of 1.6 Th, survey scans were acquired with a resolution of 120,000 at $m/z$ 200 and HCD spectra with 15,000 at $m/z$ 200.

Mass spectra were first processed with Proteome Discoverer 1.4 and then compared with a SwissProt Database with a Taxonomy filter for E coli (22,982 sequences) from 072014 using Mascot (Matrix Science, London, UK). Decoy protein sequences with a reversed order were included to estimate false discovery rates, which averaged ~1% at protein and peptide levels. Peptide spectral matches were further analyzed with Thermo Scientific Xcalibur software. Phosphoserine peptide ion intensities (within 5 ppm of calculated mass) were manually extracted, and intensity ratios of singly phosphorylated peptides relative to total (phosphorylated and nonphosphorylated) peptide intensities were calculated for each sample.

## Activity assays

Rates for ATP hydrolysis were measured with a coupled enzyme assay (*Warren et al., 1974*) with 5 µg of purified KdpFABC in a total volume of 0.5 mL at 25°C. The assay buffer contained of 75 mM TES pH 7, 150 mM KCl, 7.5 mM $MgCl_2$, 9.6 U/mL lactate dehydrogenase, 9.6 U/mL pyruvate kinase, 2.4 mM ATP, 0.5 mM phosphoenol-pyruvate, 0.36 mM NADH, and 0.15% DM. Concentrations of KCl and ATP were varied for determination of $K_M$. Orthovanadate was prepared by adjusting the pH of a 200 mM solution of sodium othovanadate to 10 using NaOH, boiling until the yellow color cleared, cooling to room temperature, and repeating this cycle until the pH stabilized. Assays were run in triplicate and data from titrations with $K^+$, ATP and $VO_4$ were fit using Prism8 software (Graph-Pad Software, San Diego, CA).

For measurement of $K^+$ transport, KdpFABC was reconstituted into proteoliposomes using the method of *Lévy et al., 1992*. Specifically, a thin film of lipid (3:1 weight ratio of *E. coli* polar extract and 1-palmitoyl-2-oleoyl-sn-glycero-3-phosphocholine) (Avanti Polar Lipids, Alabaster, AL) was prepared by evaporating a chloroform stock solution (25 mg/ml) with Ar gas followed by 2 hr in a vacuum chamber. This dry film was resuspended in transport buffer (20 mM HEPES pH 7.2, 5 mM $MgSO_4$, 140 mM potassium sulfamate, 1 mM N-methyl-D-glucamine sulfamate) to make a 10 mg/mL lipid stock solution, which was subjected to five cycles of freezing and thawing using liquid $N_2$. The stock solution was then extruded 13 times through a 400 nm nucleopore filter (Whatman plc, Maidstone, UK) to create homogenous, unilamellar liposomes. Proteoliposomes were made at a lipid:protein weight ratio of 20:1 by combining 1.25 mg liposomes, 62.5 µg of KdpFABC, and 375 µg Triton X-100 (Sigma Aldrich) in a total volume of 250 µL. This solution was stirred at room temperature for 30 min followed by addition of 7.5 mg BioBeads (BioRad) and a further 90 min incubation. Finally, 15 mg BioBeads were added before overnight incubation with stirring at 4°C. Reconstituted proteoliposomes were harvested and bath-sonicated three times for 10 s each before use in the transport assay.

The transport assay was performed as described by *Damnjanovic et al., 2013* using the voltage-sensing dye DiSC3 (Anaspec, Fremont, CA). The fluorescent signal was measured with a Fluoromax-4 spectrofluorimeter (Horiba Scientific, Piscataway, NJ) with excitation at 650 nm (5 nm slit), and emission at 675 nm (5 nm slit). Each assay was done at 4°C in 2 mL reaction volumes comprising 25 µL proteoliposomes diluted in a transport buffer consisting of 25 mM HEPES pH 7.2, 5 mM magnesium sulfamate, 140 mM potassium sulfamate and 1 mM N-methyl-D-glucamine sulfamate, followed by the addition of 1 µM DiSC3 from a 1 mM stock in DMSO. Electrogenic transport by inside-out KdpFABC molecules was initiated by addition of 2 mM ATP, and the subsequent drop in fluorescence was fit to an exponential curve in order to calculate initial rate of transport. Assays were performed in triplicate.

For measurements of EP formation, aliquots of purified KdpFABC were diluted to 0.25 mg/ml in 50 mM HEPES pH 7.8, 2 mM $MgCl_2$, 0.15% DM and 10 mM of either KCl or NaCl. The reaction was started by addition of 100 µM [$\gamma^{32}$P]-ATP and 200 µL aliquots containing 25 µg of protein and 2.7 µCi were taken at intervals of 0, 15, 30, 60, 120 s and quenched with 300 µl of 35% ice cold trichloroacetic acid (TCA). For the pulse-chase experiments, samples were incubated for 15 s at 4°C, an aliquot was recovered followed by addition of 10 mM ADP and 10 mM EDTA with additional aliquots taken various time intervals. After completing the assay, aliquots were centrifuged for 10 min at 14,000 g and washed twice with either 5% ice-cold TCA (for filtration) or water (for SDS-PAGE). For

filtration, pellets were resuspended in 500 µL 5% TCA, filtered with 0.22 mm MCE filters (Millipore-Sigma, Burlington, MA) and washed three times with 5% TCA. Radioactivity was then quantified with a scintillation counter. For SDS-PAGE, the pellets were resuspended in 30 µL of 5 mM TRIS-phosphate pH 5.8, 6.7 M urea, 0.4 M DTT, 5% SDS, 0.014% bromophenol blue. The stacking gel contained 4% polyacrylamide, 65 mM TRIS-phosphate pH 5.5, 0.1% SDS. The resolving gel contained 7.5% polyacrylamide, 65 mM TRIS-phosphate pH 6.5, 0.1% SDS. The running buffer was 0.17 M MOPS pH 6 and 0.1% SDS. Gels were dried and exposed to a storage phosphor screen overnight which was then imaged with a Typhoon Trio+ (GE Healthcare). For each construct, data was obtained from three to four independent cell expression/purification procedures and individual samples were measured in triplicate.

## Acknowledgements

We thank Jingjing Deng for initial mass spectrometry analyses. Mass spectra data has been deposited in the MassIVE repository (MSV000084906). The work was supported by NIH grants R01 GM108043 to DLS and S10 RR027990 to TAN, by funding from the European Research Council (grant agreement No. 637372) and the Independent Research Fund Denmark (grant agreement No. DFF-8021-00161) to BPP, and by Lundbeckfonden (grant No. R82-2011-7280) and Novo Nordisk Fonden (grant No. NNF18OC0034936) to HK.

## Additional information

### Funding

| Funder | Grant reference number | Author |
|---|---|---|
| National Institutes of Health | R01 GM108043 | David L Stokes |
| National Institutes of Health | S10 RR027990 | Thomas A Neubert |
| European Research Council | 637372 | Bjorn P Pedersen |
| Independent Research Fund Denmark | DFF-8021-00161 | Bjorn P Pedersen |
| Lundbeckfonden | R82-2011-7280 | Himanshu Khandelia |
| Novo Nordisk Fonden | NNF18OC0034936 | Himanshu Khandelia |

The funders had no role in study design, data collection and interpretation, or the decision to submit the work for publication.

### Author contributions

Marie E Sweet, Conceptualization, Formal analysis, Validation, Investigation, Methodology, Writing - original draft, Writing - review and editing; Xihui Zhang, Formal analysis, Investigation, Writing - review and editing; Hediye Erdjument-Bromage, Conceptualization, Formal analysis, Validation, Investigation, Methodology, Writing - review and editing; Vikas Dubey, Investigation, Methodology, Writing - review and editing; Himanshu Khandelia, Conceptualization, Supervision, Investigation, Methodology, Writing - review and editing; Thomas A Neubert, Formal analysis, Supervision, Validation, Methodology, Writing - review and editing; Bjørn P Pedersen, Conceptualization, Validation, Writing - review and editing; David L Stokes, Conceptualization, Data curation, Formal analysis, Supervision, Funding acquisition, Methodology, Writing - original draft, Project administration, Writing - review and editing

### Author ORCIDs

Bjørn P Pedersen https://orcid.org/0000-0001-7860-7230
David L Stokes https://orcid.org/0000-0001-5455-8163

### Decision letter and Author response

Decision letter https://doi.org/10.7554/eLife.55480.sa1
Author response https://doi.org/10.7554/eLife.55480.sa2

## Additional files

### Supplementary files
• Transparent reporting form

### Data availability
Raw mass spectrometry files have been deposited to the MassIVE database under accession code MSV000084906.

The following dataset was generated:

| Author(s) | Year | Dataset title | Dataset URL | Database and Identifier |
|-----------|------|---------------|-------------|-------------------------|
| Neubert TA | 2020 | Serine Phosphorylation Regulates the P-type Potassium pump KdpFABC | https://massive.ucsd.edu/ProteoSAFe/dataset.jsp?task=686cb5ce41-da4e2ea053b88212-e91285 | MassIVE, MSV0000 84906 |

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
