## [Decision Letter]

**Acceptance summary:**

This paper provides convincing evidence that post-translational modification by phosphorylation is a key feature of the bacterial potassium uptake system KdpFABC, an unusual complex of a P-type ATPase (subunit KdpB) and a potassium channel. Specifically, the results show that phosphorylation serves to inhibit the complex once cellular potassium levels have been restored, a novel regulatory strategy that may present another unique feature of KdpFABC systems, but that also provides a new lens through which other P-type ATPases should be viewed.

**Decision letter after peer review:**

Thank you for submitting your article "Serine Phosphorylation Regulates the P-type Potassium pump KdpFABC" for consideration by *eLife*. Your article has been reviewed by 3 peer reviewers, and the evaluation has been overseen by a Reviewing Editor and Olga Boudker as the Senior Editor. The following individuals involved in review of your submission have agreed to reveal their identity: Howard Young (Reviewer #3).

The reviewers have discussed the reviews with one another and the Reviewing Editor has drafted this decision to help you prepare a revised submission. Please aim to submit the revised version within two months.

Summary:

The bacterial potassium uptake system KdpFABC is an unusual complex of a P-type ATPase (subunit KdpB) and a potassium channel. Its expression levels are increased in response to low-potassium stress conditions, at which point it utilizes ATP to pump K^+^ into the cell thereby restoring the intracellular potassium concentration. Surprisingly, recent structural data for KdpFABC complexes revealed the phosphorylation of a conserved TGES motif in the A domain of KdpB, and subsequent experiments demonstrated that phosphorylation reduces the ATPase activity of KdgFABC. Using activity and phosphorylation assays, and mass spectrometry data, Sweet et al. demonstrate that this post-translation regulation occurs in response to increased K^+^ levels. Phosphorylation therefore serves to inhibit the complex once the cellular potassium levels have been restored. This novel regulatory strategy may present another unique feature of KdpFABC systems, but also provides a new lens through which other P-type ATPase should be viewed.

Essential revisions:

While the observations regarding the role of Ser162 in phosphorylation are clear and robust, more conclusive data must be provided regarding the molecular basis of inhibition, according to the following:

1) The results describing the experiments identifying the step in the Post-Alber cycle affected by phosphorylation are confusing, in part because of the various mutants and growth conditions. This section requires clarification. It may also help to rename E161Q to reflect the additional mutations that accompany it.

2) Based on Figure 4 it is concluded that KdpFABC is inhibited in the E1~P state. There are two issues with the experiment.

- First, all variants were produced at high K, and thus in the Q116R mutant, Ser162 would be mostly phosphorylated. The authors claim that the phosphorylation of Asp307 is uncoupled from the presence of potassium, which is why high phosphorylation is found for the inhibited variants Q116R and S162D (Figure 4A). But why does the EP level of Q116R decrease over time, if it is inhibited? The "rate" of dephosphorylation is at least as fast as for S162A, which is not inhibited and the dephosphorylation is nearly 100%. Considering that only a fraction of the protein (e.g. 75%) is inactive, the dephosphorylation should only apply to a small portion of the phosphorylated protein. How does this experiment look for Q116R expressed at low K? Is the initial EP level much lower due to fast turnover as suggested for S162A? And is the dequenching even faster?

- Second, from the pulsed-chase experiment the authors claim that the protein is trapped in an E1~P state. However, from the inhibition with orthovanadate the authors concluded that the equilibrium between E1/E2 is not affected by the phosphorylation. In the best case this statement is only a bit misleading as it only refers to the non-phosphorylated fraction. If something else was meant, this statement should be clarified. Further, the pulsed-chase experiment can be explained by two scenarios: the inhibition of the protein in one state and the E1-E2 transition as rate-limiting step. For S162A the authors claim the experiment reflects the latter, while S162D is inhibited in E1-~P.

3) The plots in 2G and 2H are misleading as they are based on the assumption that upon 100% phosphorylation the ATPase activity is completely abolished. However, there is no formal proof of this. In fact, the maximal inhibition seen is approx. 60% after 90 min K shock when compared to the dephosphorylated sample. On the other hand, the level of phosphorylation cannot be reliably estimated, neither from mass spec nor from the phos-tag signal. One could thus speculate that the rate of phosphorylation is even higher, particularly because no significant increase was detected upon K shock after 20 minutes (Figure 2F), and that the phosphorylation does not completely inhibit KdpFABC but instead slows down the state transitions (e.g. E1 to E2) such that 100% inhibition would never be seen. This scenario may also explain the experiments performed in Figure 4.

4) The molecular dynamics simulations presented rely on the deletion of the phosphate group from Ser162 as a proxy for the dephosphorylated state. This assumption is flawed as it involves artifactual, high-energy states, and does not consider the metastable conformations that are likely to be involved immediately before or after kinase/phosphatase activity. Moreover, the simulations of the phosphorylated structure were initiated using a system in which the cavities in the structure have not been hydrated (e.g., using Dowser), with the exception of one highly-ordered water molecule observed in the structure. Such cavities are unlikely to become hydrated on the timescale of the equilibration, but could have a significant effect on the dynamics of the protein system. Therefore, these simulation data should be removed. In case the authors chose to provide new computational data in its place, substantially more detail must be provided to describe the analysis, e.g. which atoms are included in RMSD calculations. Please also note that radial distribution functions are not typically appropriate for describing residue-residue interactions; distance distributions are preferable.

5) A central theme of the proposed mechanism is that Ser162-Pi interacts with K357+R363, as observed in the available crystal structure (PDB code 5MRW) and that dephosphorylation would abolish this interaction leading to an allosteric effect on the transmembrane domains. However, other structures of KdpFABC (PDB codes 6HRA and 6HRB; from Stock et al.,) contain Ser162-Pi and yet do not show this interaction, despite representing, at least in one case, a similar state in the transport cycle. Moreover, one of those structures is an E2 state of Ser-phosphorylated KdpB, which questions the hypothesis that the phosphorylated state is truly inhibited in an E1 conformation. Thus, any discussion of a molecular mechanism must carefully and convincingly consider those structures and their implications. [We note that the lower resolution of the cytoplasmic domains in the cryo-EM structure obtained for the E1 state point towards a higher flexibility of this region (supporting the conclusion drawn here that phosphorylation increases the mobility). Yet, despite the lower resolution the overall position of the A, N and P domains is unambiguous enough to show that these contacts are disrupted.]

[Editors' note: further revisions were suggested prior to acceptance, as described below.]

Thank you for submitting your article "Serine Phosphorylation Regulates the P-type Potassium pump KdpFABC" for consideration by *eLife*. Your article has been reviewed by Olga Boudker as the Senior Editor, a Reviewing Editor, and two reviewers. The reviewers have opted to remain anonymous. The Reviewing Editor has also reviewed the article, and has drafted this decision to help you prepare a revised submission.

Summary:

The bacterial potassium uptake system KdpFABC is an unusual complex of a P-type ATPase (subunit KdpB) and a potassium channel. Its expression levels are increased in response to low-potassium stress conditions, at which point it utilizes ATP to pump K^+^ into the cell thereby restoring the intracellular potassium concentration. Recently, structural data for KdpFABC complexes revealed a surprising observation, namely phosphorylation of a serine in a conserved TGES motif in the A domain of KdpB, based on which it was shown that phosphorylation reduces the ATPase activity of KdgFABC. In the present work, Sweet et al., demonstrate that this post-translation regulation occurs in response to increased K^+^ levels, using activity and phosphorylation assays, and mass spectrometry data. Phosphorylation therefore serves to inhibit the complex once the cellular potassium levels have been restored. This novel regulatory strategy may present another unique feature of KdpFABC systems, but also provides a new lens through which other P-type ATPases should be viewed.

The authors satisfactorily addressed the major concerns regarding the biochemical aspects of the work, and the experimental systems and major conclusions of the work are more clearly presented. They also provided additional experimentation that test a foundational assumption regarding the simulation data. Unfortunately, the new experiments illustrate that key residues interacting by the phosphorylserine group in the starting structure of the MD simulations are not required for inhibition. Therefore, the simulations of this structure are unlikely to be relevant for the (very challenging) question at hand. Moreover, the authors use, as their control, a modified (dephosphorylated) version of that structure, which is bound to have unfavorable interactions that most probably do not reflect the true dephosphorylated protein. [On this latter point, based on the authors responses it is apparent that one of the comments in the first round of reviews was insufficiently clear, and therefore, we provide a clarification below.] Based on these concerns, if the authors wish to resubmit a revised version of the manuscript, the MD simulations must be removed.

Clarification:

In referring to a high-energy, artifactual state, we referred not to the crystal structures themselves, but rather to the molecular system created by the authors when the phosphate group was deleted from the serine in the crystal structure prior to running the MD simulations, i.e., the SER system. The concern is that introducing this dramatic artificial change into a structure (in this case, one in which the phosphoryl group interacts with two positively-charged residues) creates an system with a set of new high-energy interactions; these could be the cause of the "uncoupling of the A- and N-domains" (subsection “Effects of serine phosphorylation on structural dynamics”). In other words, the SER system does not reflect a true dephosphorylated protein. The true dephosphorylated protein may or may not ever visit the exact same conformation that is present in this starting structure. In an ideal world, one would initiate the simulations with an experimental structure of a dephosphorylated protein in the same state of the transport cycle. Obviously, such a structure is not available, but the alternative adopted here is insufficiently robust to address the question at hand.

---

## [Author Response]

Essential revisions:While the observations regarding the role of Ser162 in phosphorylation are clear and robust, more conclusive data must be provided regarding the molecular basis of inhibition, according to the following:1) The results describing the experiments identifying the step in the Post-Alber cycle affected by phosphorylation are confusing, in part because of the various mutants and growth conditions. This section requires clarification. It may also help to rename E161Q to reflect the additional mutations that accompany it.

We agree that the diversity of samples used for this work could cause confusion. Keeping track of the wide array of mutations and varying growth conditions is critical for understanding the implications of individual experiments and for the overall functioning of KdpFABC. To assist in this task, we have added a new paragraph at the beginning of the Results section to enumerate the mutations and to explain differences in the growth conditions used to produce them. This explanation is accompanied by a newly added Figure 2, which diagrammatically illustrates the bacterial strains as well as K^+^ concentrations used for growth, as well as Table 1, which summarizes information for each construct. We have also changed the nomenclature for the constructs in the text to explicitly acknowledge double and triple mutations.

2) Based on Figure 4 it is concluded that KdpFABC is inhibited in the E1~P state. There are two issues with the experiment.

Note that data for these experiments are now shown in Figure 5 of the revised manuscript.

- First, all variants were produced at high K, and thus in the Q116R mutant, Ser162 would be mostly phosphorylated. The authors claim that the phosphorylation of Asp307 is uncoupled from the presence of potassium, which is why high phosphorylation is found for the inhibited variants Q116R and S162D (Figure 4A). But why does the EP level of Q116R decrease over time, if it is inhibited? The "rate" of dephosphorylation is at least as fast as for S162A, which is not inhibited and the dephosphorylation is nearly 100%. Considering that only a fraction of the protein (e.g. 75%) is inactive, the dephosphorylation should only apply to a small portion of the phosphorylated protein.

We do have an explanation for the decrease in EP levels for the Q116R mutant. As the reviewers acknowledge, this preparation has a mixture of serine phosphorylated (inhibited) and unmodified (fully active) protein. We believe that the fraction that is serine phosphorylated is blocked in the E1~P conformation, whereas the active fraction will turnover. As a result of this enzymatic turnover, ADP will be produced and, as demonstrated by the pulse-chase experiments (Figure 5D), ADP rapidly reacts with the E1~P species to reverse the phosphorylation of Asp307 (step 2 in Figure 1). Thus, as time goes on and more ADP is produced, the total steady-state levels of EP will fall to zero. This effect only occurs in the presence of K, because otherwise there is no turnover and no ADP is produced. We previously thought that this detail might be confusing to readers, but we are pleased that the reviewers brought it up and we have included an explanation in subsection “Effect of serine phosphorylation on partial reactions” of the revised manuscript.

How does this experiment look for Q116R expressed at low K? Is the initial EP level much lower due to fast turnover as suggested for S162A? And is the dequenching even faster?

With our current expression strategies, we are not able to produce the Q116R mutant at low K. Because of the lower K affinity produced by this mutation, it cannot rescue cell growth at μM levels of K, but requires at least 0.2 mM at which concentration there is a already a substantial amount of serine phosphorylation. In principle, one could try to express a His-tagged Q116R construct in a background of WT KdpFABC. Although we do not have the strain necessary to attempt this experiment, it would be interesting to pursue this approach in follow-up studies.

- Second, from the pulsed-chase experiment the authors claim that the protein is trapped in an E1~P state. However, from the inhibition with orthovanadate the authors concluded that the equilibrium between E1/E2 is not affected by the phosphorylation. In the best case this statement is only a bit misleading as it only refers to the non-phosphorylated fraction. If something else was meant, this statement should be clarified. Further, the pulsed-chase experiment can be explained by two scenarios: the inhibition of the protein in one state and the E1-E2 transition as rate-limiting step. For S162A the authors claim the experiment reflects the latter, while S162D is inhibited in E1-~P.

The titrations with orthovandate, K and ATP address the possibility that serine phosphorylation alters the rate of a step in the reaction cycle rather than blocking it altogether. If rates were affected, the fraction of unphosphorylated molecules would behave normally, but the serine-phosphorylated molecules – comprising the majority of the population – would behave differently and thus shift one of these titration curves. The fact that all three titration curves, now shown in Figure 6, were not significantly affected by levels of serine phosphorylation is consistent with (though not necessarily proof of) our hypothesis that serine phosphorylation completely blocks turnover and that the observed activity comes only from unphosphorylated molecules. In order to make this reasoning clearer, we have moved this section to a later point in the manuscript so that there is better context for these experiments. We have also rewritten the relevant paragraphs in subsection “Effect of serine phosphorylation on ligand binding”.

3) The plots in 2G and 2H are misleading as they are based on the assumption that upon 100% phosphorylation the ATPase activity is completely abolished. However, there is no formal proof of this. In fact, the maximal inhibition seen is approx. 60% after 90 min K shock when compared to the dephosphorylated sample. On the other hand, the level of phosphorylation cannot be reliably estimated, neither from mass spec nor from the phos-tag signal. One could thus speculate that the rate of phosphorylation is even higher, particularly because no significant increase was detected upon K shock after 20 minutes (Figure 2F), and that the phosphorylation does not completely inhibit KdpFABC but instead slows down the state transitions (e.g. E1 to E2) such that 100% inhibition would never be seen. This scenario may also explain the experiments performed in Figure 4.

As discussed above for point 2, the data are consistent with the idea that serine phosphorylation completely inhibits turnover. This conclusion is further supported by results from the S162D mutation, which is completely inactive and produces a steady level of EP in both the presence and absence of K. However, the reviewers are correct that we have no formal proof that serine phosphorylation has this same effect and perhaps S162D is a misleading – though commonly used – mimetic of phosphorylation. The right-hand axes are not essential to our story so we have removed them from the revised figures and will simply report phosphorylation in arbitrary units, since this will convey the point without unnecessary confusion or controversy.

The data after 90 min of K shock is somewhat puzzling, as one would logically expect continued increases in inhibition over time. After reconsidering this matter, however, we believe that cell metabolism is likely to be compromised under extended conditions of K shock. Unlike the complete inhibition observed in the previous publication by Roe et al., (2000), the inhibitory system (as yet uncharacterized) is not 100% effective in shutting down KdpFABC activity in our expression system. In the original manuscript, we speculated that this outcome is due to forced overexpression of KdpFABC from a multicopy plasmid that has overwhelmed this regulatory system. As a result of this deficiency, we expect that K shock will be detrimental to ionic homeostasis and potentially compromise survival of cells after they are returned to K replete conditions. If so, K shock would cause unexpected results during long incubations This effect may be reflected in the lower activity seen in LPP treated samples from the longer periods of K shock. This issue is discussed in more detail in the Discussion section in the revised manuscript.

4) The molecular dynamics simulations presented rely on the deletion of the phosphate group from Ser162 as a proxy for the dephosphorylated state. This assumption is flawed as it involves artifactual, high-energy states, and does not consider the metastable conformations that are likely to be involved immediately before or after kinase/phosphatase activity.

The reviewers raise important points about our analysis of MD simulations that have caused us to reconsider their contribution. The bottom line is that simulations show a dramatic difference in the behavior of serine phosphorylated and unphosphorylated KdpFABC and, although they do not offer definitive answers, we believe that they make a valuable contribution to our story. For the revised manuscript, we have refocused the analysis on the dynamic properties of cytoplasmic domains of KdpB and on the linkers connecting the A-domain to the membrane helices. We have downplayed the issue of hydration and have presented a testable hypothesis to explain the functional effects seen in our biochemical experiments.

We are not certain what is meant by "artifactual high-energy states" and the "metastable conformations […] immediately before or after kinase/phosphatase activity". In our minds, this statement could refer to either (1) states related to the conformational cycle of the P-type pump and phosphorylation/hydrolysis of Asp307, (2) the regulatory process involving phosphorylation/dephosphorylation of the Serine162 or (3) the influence of crystal environment that could force flexible regions to adopt a high-energy conformation. For our response, we have assumed that the "artifactual high-energy state" refers to the packed environment in the crystals that produced the starting model for our MD simulations (#3 above) and that kinase/phosphatase activity refers to the unknown agent that is acting on Ser162 (#2 above). We sincerely hope that this is as intended.

We agree that the X-ray structure is possibly in a high-energy conformation, stabilized by the chemical environment of the crystallization experiment (crystal packing, chemical composition of reservoir solution etc.). We nevertheless used the X-ray structure because it is based on data of considerable higher quality compared to the EM models. This is exemplified by the higher resolution of the X-ray data (2.9 Å) compared to the cryo-EM structures (3.7 and 4.0 Å), and is especially evident in the definition of the cytoplasmic domains (cryo-EM structures have local resolution of 5-6 Å in this region). We note that crystal packing is likely not the primary determinant of the observed conformation, given that the asymmetric unit contained three copies of KdpFABC, each having unique interactions with neighboring molecules but all adopting the same conformation (suitable for non-crystallographic symmetry averaging). In general, the use of crystal structures as a starting point of Molecular Dynamics is standard practice, and any "high-energy" strain imposed by the crystallographic experiment is normally relieved either by the initial equilibration procedure or during the restraint-free production simulations. It is the undeniable that the choice of the starting conformation can potentially influence the average thermodynamic quantities calculated from the simulations, but we feel that our choice is well justified and was effective not only in revealing distinct character for the two systems but also in highlighting a region of the molecule that may be responsible for allosteric coupling. For the revised manuscript, we added a brief explanation for the basis of selecting the X-ray structure over the cryo-EM structures in subsection “Effects of serine phosphorylation on structural dynamics”.

With regard to the metastable intermediate states associated with phosphorylation, it was not our goal to study this process, but rather to compare the properties of the protein in the presence and absence of serine phosphorylation.

Moreover, the simulations of the phosphorylated structure were initiated using a system in which the cavities in the structure have not been hydrated (e.g., using Dowser), with the exception of one highly-ordered water molecule observed in the structure. Such cavities are unlikely to become hydrated on the timescale of the equilibration, but could have a significant effect on the dynamics of the protein system. Therefore, these simulation data should be removed. In case the authors chose to provide new computational data in its place, substantially more detail must be provided to describe the analysis, e.g. which atoms are included in RMSD calculations.

There is ample evidence in literature that the free water movement in MD simulations is able to accurately hydrate protein cavities. For example, Tajkhorshid and colleages showed that cavities of amino acid transporters were hydrated on a timescale comparable to our simulations (~100 to 200 ns) (Table 1 in Li et al., 2013). In another example, water molecules spontaneously hydrate or dehydrate a key enzymatic active site in the cytochrome c oxidase (Goyal et al., 2013). With regard to ion pumps, we have not been able to find any report on MD simulations in which a hydrating procedure was used (Rui et al., 2016; Ratheal et al., 2010; Sugita et al., 2010; Espinoza-fonseca et al., 2015; Sonntag et al., 2011, and several of our own investigations: Yamamoto et al., 2019; Dubey et al., 2018; Mahmmoud et al., 2015; Poulsen et al., 2010). These examples illustrate the broad acceptance of MD simulations of transmembrane protein systems without using cavity-hydrating programs such as Dowser. As a result, we believe that the distribution of water seen during our simulations has good precedent in the field as a measure of hydration for KdpFABC.

We did nevertheless use Dowser++ as proposed on the KdpFABC as well as on the Na/K-ATPase system that we have previously examined in a number of publications. During this process, we communicated with the authors of Dowser++ to ensure that our protocol was optimal (specifically that a draining process was applied after the initial hydration). Despite our best effort, we obtained gross over-hydration of the membrane domain for both protein systems. As a result of this over-hydration, the ensuing MD runs produced anomalous results. In reviewing the use of Dowser++ in other publications (Farahvash et al., 2018, Morozenko et al., 2016), we noted that the number of water molecules inserted by Dowser++ often increases significantly as the resolution of the structure decreases and that very high-resolution (sub-Å) crystal structures are typically used for benchmarking Dowser++ performance.

Given the uncertainties of using Dowser++ with low resolution structures, and the poor behavior of the resulting hydrated complexes of KdpFABC and Na/K-ATPase during simulations, we believe that this issue requires further experimentation which is outside of the scope of the current manuscript. We have therefore refocused the revised manuscript on the dynamics of the cytoplasmic domains and the identification of a potential allosteric control site in the linker between the A-domain and the membrane domain. Although our observation of increased hydration is noted as a potential explanation for the K^+^ independence of EP formation caused by serine phosphorylation, we have de-emphasized this point by moving it to the Discussion section with the relevant data moved to the supplemental material (Figure 9—figure supplement 1). We believe that this observation provides a potentially important lead for future studies into the allosteric effects of serine phosphorylation and therefore makes a valuable contribution to the manuscript.

Please also note that radial distribution functions are not typically appropriate for describing residue-residue interactions; distance distributions are preferable.

We agree that distance distributions may be more suited to represent residue-residue interactions. We have therefore included RMSD distance distributions alongside the radial distribution functions. In general, RMSD was calculated from the last 100 ns of the simulations after correcting individual structures for bulk translational motion; the calculations include all atoms from the relevant residues. The radial distribution functions and RMSD values both show comparable effects and support the striking difference in behavior after phosphorylation of Ser162. The details of the RMSD calculation have been included in legend for Figure 7—figure supplement 1.

5) A central theme of the proposed mechanism is that Ser162-Pi interacts with K357+R363, as observed in the available crystal structure (PDB code 5MRW) and that dephosphorylation would abolish this interaction leading to an allosteric effect on the transmembrane domains. However, other structures of KdpFABC (PDB codes 6HRA and 6HRB; from Stock et al) contain Ser162-Pi and yet do not show this interaction, despite representing, at least in one case, a similar state in the transport cycle. Moreover, one of those structures is an E2 state of Ser-phosphorylated KdpB, which questions the hypothesis that the phosphorylated state is truly inhibited in an E1 conformation. Thus, any discussion of a molecular mechanism must carefully and convincingly consider those structures and their implications. [We note that the lower resolution of the cytoplasmic domains in the cryo-EM structure obtained for the E1 state point towards a higher flexibility of this region (supporting the conclusion drawn here that phosphorylation increases the mobility). Yet, despite the lower resolution the overall position of the A, N and P domains is unambiguous enough to show that these contacts are disrupted.]

These are very valid observations that indicate our failure to express clearly our point of view about the interaction between Ser162-Pi and K357-R363. Although this interaction was highlighted in the X-ray structure and proposed as a potential mechanism for inhibition, the paper also noted lack of conservation for K357 and R363 in KdpB from other species. Given the cryo-EM structures from the Hänelt group and the MD studies presented here, our view on this matter has shifted. We now believe that the observed interaction with K357 and R363 in the crystal structure is not essential for the inhibition to occur. We have brought this issue front-and-center in the Introduction of the revised manuscript (Introduction) and return to the lack of sequence conservation in the Discussion section. In addition, we have added a key experiment to show that mutation of Lys357 and Arg363 does not prevent inhibition (Figure 7—figure supplement 1G and H). In particular, we expressed the K357A/R363A/Q116R mutant in the TK2498 strain in 0.2 mM KCl (same conditions as for Figure 3) and used the Phos-tag stain to show that serine phosphorylation had occurred. We then compared ATPase activity of this preparation before and after LPP treatment. The ~4-fold stimulation produced by the phosphatase is analogous to the effect on WT protein, providing clear evidence that Lys357 and Arg363 are not essential for the inhibitory mechanism. These new data are referenced together with the MD results (subsection “Effects of serine phosphorylation on structural dynamics”).

With regard to the cryo-EM structures from the Hänelt group, these two structures were derived from a single sample prepared in the presence of AMP-PCP, K, and AlF_4_. Given our evidence that serine-phosphorylation eliminates K-dependence of the reaction cycle, this combination of ligands could plausibly stabilize E1, E1-ATP, E1~P, or E2-P states. Although we conclude that serine-phosphorylation blocks the reaction cycle in the E1~P state, the E2-P state would still be accessible via back-door phosphorylation, especially given the uncoupling of the Post-Albers reaction cycle with respect to K. These thoughts about the cryo-EM structures are included in the penultimate paragraph of the Discussion section, which also dismisses roles for Lys357 and Arg363 and offers an alternative explanation for stabilization of the E1~P state.